# The impact of surgery and oncological treatment on risk of type 2 diabetes onset in patients with colorectal cancer: nationwide cohort study in Denmark

Caroline Krag[1,2†], Maria Saur Svane[1,3†], Sten Madsbad[1], Susanne Boel Graversen[4], Jesper Frank Christensen[5,6,7], Thorkild IA Sørensen[8], Louise Lang Lehrskov[5*†], Tinne Laurberg[4†]

[1]Department of Endocrinology, Copenhagen University Hospital Hvidovre, Hvidovre, Denmark; [2]Department of Medicine, Faculty of Health and Medical Sciences, University of Copenhagen, Copenhagen, Denmark; [3]Department of Surgical Gastroenterology, Copenhagen University Hospital Hvidovre, Hvidovre, Denmark; [4]Steno Diabetes Center Aarhus, Aarhus University Hospital, Aarhus, Denmark; [5]Centre for Physical Activity Research, Copenhagen University Hospital - Rigshospitalet, Copenhagen, Denmark; [6]Department of Sports Science and Clinical Biomechanics, Faculty of Health Sciences, University of Southern Denmark, Odense, Denmark; [7]Digestive Disease Center, Bispebjerg Hospital, Copenhagen, Denmark; [8]Novo Nordisk Foundation Center for Basic Metabolic Research and Department of Public Health, University of Copenhagen, Copenhagen, Denmark

*For correspondence: louise.lang.lehrskov.01@regionh.dk

†These authors contributed equally to this work

## Abstract

**Background:** Comorbidity with type 2 diabetes (T2D) results in worsening of cancer-specific and overall prognosis in colorectal cancer (CRC) patients. The treatment of CRC per se may be diabetogenic. We assessed the impact of different types of surgical cancer resections and oncological treatment on risk of T2D development in CRC patients.

**Methods:** We developed a population-based cohort study including all Danish CRC patients, who had undergone CRC surgery between 2001 and 2018. Using nationwide register data, we identified and followed patients from date of surgery and until new onset of T2D, death, or end of follow-up.

**Results:** In total, 46,373 CRC patients were included and divided into six groups according to type of surgical resection: 10,566 Right-No-Chemo (23%), 4645 Right-Chemo (10%), 10,151 Left-No-Chemo (22%), 5257 Left-Chemo (11%), 9618 Rectal-No-Chemo (21%), and 6136 Rectal-Chemo (13%). During 245,466 person-years of follow-up, 2556 patients developed T2D. The incidence rate (IR) of T2D was highest in the Left-Chemo group 11.3 (95% CI: 10.4–12.2) per 1000 person-years and lowest in the Rectal-No-Chemo group 9.6 (95% CI: 8.8–10.4). Between-group unadjusted hazard ratio (HR) of developing T2D was similar and non-significant. In the adjusted analysis, Rectal-No-Chemo was associated with lower T2D risk (HR 0.86 [95% CI 0.75–0.98]) compared to Right-No-Chemo.

For all six groups, an increased level of body mass index (BMI) resulted in a nearly twofold increased risk of developing T2D.

**Conclusions:** This study suggests that postoperative T2D screening should be prioritised in CRC survivors with overweight/obesity regardless of type of CRC treatment applied.

**Funding:** The Novo Nordisk Foundation (*NNF17SA0031406*); TrygFonden (101390; 20045; 125132).

### eLife assessment

This **valuable** study presents findings that suggest the need for postoperative type 2 diabetes screening and that this should be prioritized in colorectal cancer survivors with overweight/obesity regardless of the type of colorectal cancer treatment applied. The evidence supporting the claims of the authors is **solid** and the authors use a population-based cohort study including all Danish colorectal patients who had undergone colorectal cancer surgery between 2001-2018. The work will be of interest to medical biologists, endocrinologists and oncologists working on colorectal cancer.

## Introduction

Colorectal cancer (CRC) is the third most common cancer worldwide (*Araghi et al., 2019*), and the incidence rate (IR) is still increasing (*Arnold et al., 2017*). The same applies to the global IR of type 2 diabetes (T2D). The association between CRC and T2D is well established, as patients with T2D have approximately 30% increased risk of developing CRC (*González et al., 2017*), and T2D is known to negatively influence the prognosis of CRC (*Mills et al., 2013*). Furthermore, some studies even show an increased T2D risk in CRC survivors (*Singh et al., 2016*; *Xiao et al., 2021*).

Fortunately, survival after a CRC diagnosis is steadily improving due to early detection, diagnostic precision, and more advanced cancer treatment; and today, more than 70% of non-metastatic CRC patients are alive after 5 years (*Colorectal, 2019*). Accordingly, it becomes increasingly important to identify CRC patients at risk of developing long-term health problems like T2D in order to initiate preventive strategies or treatment (*Xiao et al., 2021*). The underlying mechanisms behind the increased T2D risk in CRC survivors are poorly understood. However, shared lifestyle-related risk factors such as obesity, physical inactivity, and unhealthy diet may play a role (*Nano et al., 2020*; *Maddatu et al., 2017*; *Lega and Lipscombe, 2020*).

Another explanation may be treatment-induced hyperglycemia. Prior studies showed that treatment with adjuvant chemotherapy can induce hyperglycemia and development of T2D in CRC patients (*Xiao et al., 2021*; *Jo et al., 2021*; *Feng et al., 2013*). Moreover, glucocorticoid, administered to alleviate side effects of chemotherapy, may impair glucose homeostasis and thus increase the risk of T2D (*Lee et al., 2021*; *Jeong et al., 2016*). In addition to the drug-induced metabolic disturbances, a recent study revealed that the type of CRC surgery per se is associated with diverging risk of T2D (*Jensen et al., 2018*). A previous nationwide Danish study revealed that patients who had undergone left-sided colon resection for CRC and other colonic illnesses had an increased risk of developing T2D, whereas this risk was unaltered in patients who had undergone right-sided colonic resections compared with a control group (*Jensen et al., 2018*). However, this study lacks essential information on the role of potential modifying factors like body mass index (BMI), treatment with chemotherapy, etc. Along the same line, in a population of non-CRC patients, Lee et al. published that patients who had undergone right-sided or transversal colonic resections had a reduced risk of developing T2D compared with a matched non-colectomy control group (*Wu et al., 2021*). These findings raise an intriguing question of whether the risk of developing T2D in CRC survivors may be casually affected by the type of CRC surgery. Possible explanations of this side-specific risk modification include removal of different enteroendocrine cell types resulting in gut changes that is important for insulin secretion and action and for appetite regulation as well as changes in the microbiome, but causality remains to be investigated further (*Holst et al., 2016*; *Lin et al., 2019*; *Palnaes Hansen et al., 1997*).

In the present study, we tested the hypothesis whether different types of surgical cancer resection and oncological treatment have an impact on risk of developing diabetes in CRC survivors. For this purpose, we used a nationwide register to compare the frequency of diabetes development in CRC patients who had undergone right-sided colectomy, left-sided colectomy, or rectal resection with and without chemotherapy, when adjusting for potential modifying factors like BMI, etc.

## Materials and methods

### Study population

The study was based on information from Danish nationwide health registers. All data are recorded with reference to a civil registration number, which is a unique personal identification number assigned

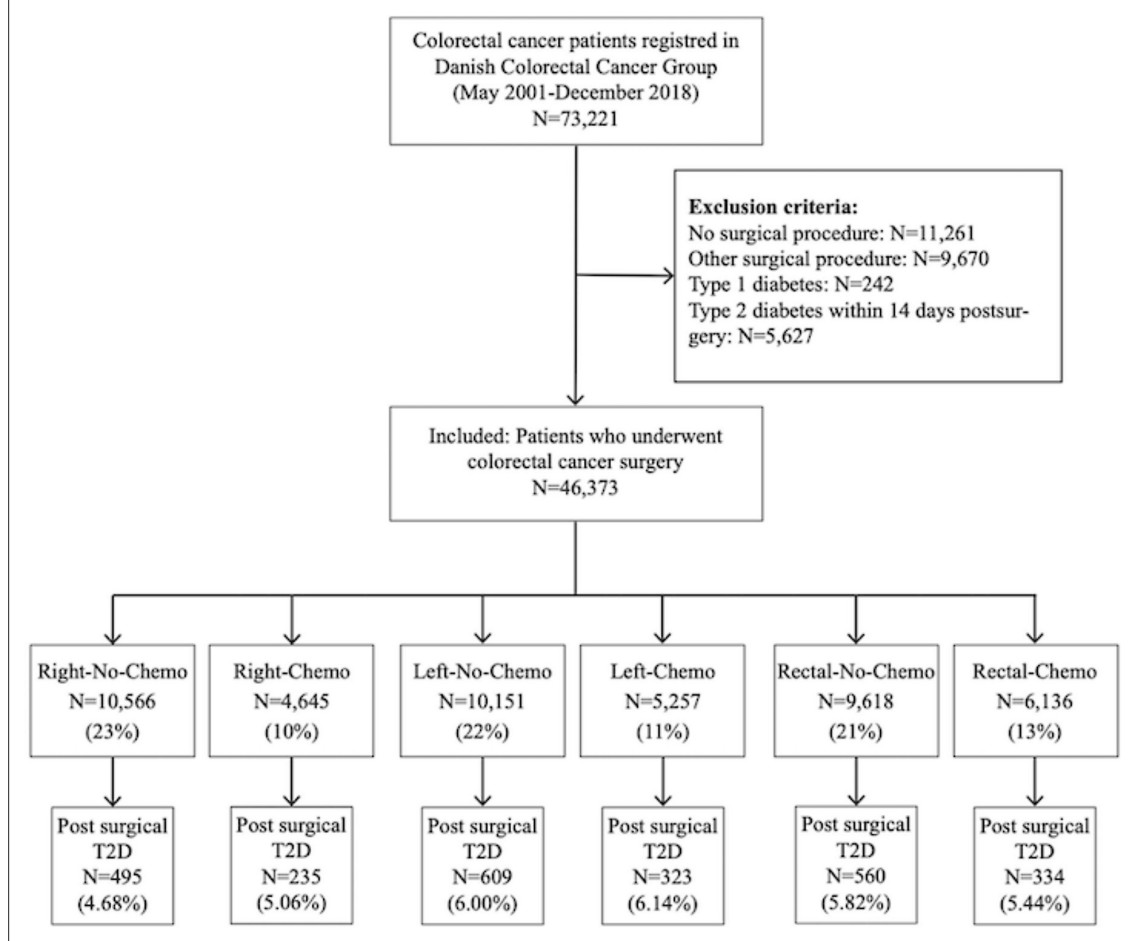

**Figure 1.** Flowchart of the study.

to all Danish residents. This number permits accurate linkage of recorded information at the personal level (*Pedersen et al., 2011*).

The study population was identified through the Danish Colorectal Cancer Group (DCCG) database, which contains prospectively collected and validated data on all patients with CRC since 2001, including data on patient's characteristics, diagnostics, and treatment (*Klein et al., 2020*). The degree of patient inclusion in the database is high (>95%) and thus adequately represents the population of Danish CRC patients undergoing cancer resections (*Klein et al., 2020*).

We included all CRC cancer patients undergoing cancer surgery between 1 May 2001 until 31 December 2018 (*Figure 1*). Patients with unspecified surgery or total colectomy were excluded as these procedures are not standard treatment for CRC. Additionally, individuals diagnosed with type 1 diabetes (T1D) either before or after surgery were excluded, along with those diagnosed with T2D preoperatively or within the first 2 weeks postoperatively, as the last group probably represents patients with preoperatively unknown pre-existing prediabetes or diabetes.

## Exposure

Patients were subdivided into six groups according to the surgical procedure and whether the patients had received chemotherapy or not. The group 'right-sided colonic resection' (Right) included patients with cancer in coecum, ascending colon, or the oral part of colon transversum, the group 'left-sided colonic resection' (Left) included patients with cancer in the anal part of colon transversum, descending colon, and sigmoid colon, and the group 'rectal resection' (Rectal) included patients with rectal cancer. Information regarding chemotherapy was limited to whether the patient had received chemotherapy (Chemo) or not (No-Chemo).

Based on the combination of surgical procedures and chemotherapy, the study population was divided into six groups: Right-No-Chemo, Right-Chemo, Left-No-Chemo, Left-Chemo, Rectal-No-Chemo, Rectal-Chemo.

Radiation therapy is only part of standard treatment for rectal cancer, and therefore this oncological modality was restricted to Rectal patients (Rectal-No-Radiation vs. Rectal-Radiation).

Finally, a sub-analysis was made comparing all colonic resected patients (All Colon) with all rectal resected (All Rectal).

## Outcome

The primary outcome was first-time diagnosis of T2D after different types of CRC cancer surgery and chemotherapy. Patients were followed from date of surgery until time to T2D diagnosis, emigration, death, or the end of follow-up (31 December 2018), whichever came first.

Individuals with T2D were identified from data in the Danish National Patient Register (comprising information on diagnoses during all hospital contacts in Denmark), the National Prescription Registry (comprising information on all prescriptions filled at any pharmacy in Denmark), and the Danish National Health Service Register (comprising information on podiatrist services) as described by *Carstensen et al., 2011*. Diabetes is defined as the second occurrence of any event across three types of inclusion events: (1) Diabetes diagnosed during hospitalisation, (2) diabetes-specific services received at podiatrist, and (3) purchases of glucose lowering drugs. Thus, if a patient developed transient T2D during chemotherapy treatment, it will only be an inclusion event if they purchase of glucose lowering drugs. Individuals were classified as having T1D if they had received prescriptions for insulin combined with a diagnosis of type 1 from a medical hospital department. Otherwise, diabetes was classified as type 2 (*Isaksen et al., 2023*).

## Statistical analysis

Descriptive data was presented as means with standard deviations (SD) for continuous variables and proportions (n, %) for categorical variables. Crude IR (number of T2D diagnoses by person-time at risk) were calculated for the six groups (type of surgery±chemo), and characterised by preoperative variables (sex, age, BMI, smoking, alcohol status, American Society of Anesthesiologists physical status classification score [ASA], and treatment with radiotherapy in rectal cancer patients). Unadjusted and adjusted Cox regression analyses were used to calculate hazard ratios (HRs) for developing T2D after different types of surgery with and without chemotherapy (using Right-No-Chemo as a reference group), after radiation therapy among Rectal resected (using Rectal-No-Radiation as reference), and after overall type of surgery (All Colon [ref] vs All Rectal). The adjusted analyses were performed stepwise by first adjusting for year of surgery, age at surgery, sex, and BMI (Model 1), then further adjusting ASA, smoking, and alcohol consumption (Model 2a), and lastly, further adjusting for chemotherapy (Model 2b) in the analyses on radiation therapy and overall type of surgery.

Finally, the impact of BMI on the risk of developing of T2D was investigated within each of the six groups. BMI was categorised into four subgroups (18.5–24.9, 25–29.9, 30–34.9, and 35–39.9), and underweight (BMI<18.5) or severely obese (BMI≥40) subjects were excluded from these analyses due to low numbers. For each group (type of surgery±chemotherapy), the HR for developing T2D depending on BMI subgroups was calculated by using Cox regression analysis adjusted for age, sex, year of surgery, and ASA score using normal weight (BMI: 18.5–24.9) as the reference group.

A sensitivity analysis was performed by restricting the inclusion period of surgery patients from 2001–2018 to 2010–2018, as baseline information was complete from 2010.

All analyses were conducted using STATA version 16 (STATA Corp., College Station, TX, USA). p-Value of p<0.05 was considered statistically significant. Results were presented with 95% confidence interval (CI) when relevant.

## Results

### Baseline characteristics

A total of 46,373 CRC patients without preoperatively recorded diabetes were found eligible for the study (*Figure 1*) and were divided into six groups: 10,566 Right-No-Chemo (23%), 4645 Right-Chemo

(10%), 10,151 Left-No-Chemo (22%), 5257 Left-Chemo (11%), 9618 Rectal-No-Chemo (21%), and 6136 Rectal-Chemo (13%).

Compared with the Right, the Left and Rectal groups were younger, more often men, and less often classified as ASA>II (*Table 1*). Those that had received chemotherapy were generally younger, less often classified as ASA>II, and as expected more often stage III/IV cancers than No-Chemo.

### IR of diabetes

During the 245,466 person-years of follow up, 2556 cases of T2D were diagnosed (10.4 per 1000 person-years) (*Table 2*) and the mean time to onset of T2D post-surgery spanned from 4.1 to 4.7 years across the subgroups. The IR of T2D was almost similar between the six groups with the highest IR observed in the Left-No-Chemo group (IR 11.3 [95% CI: 10.4–12.2] per 1000 person-years), and the lowest IR observed in the Rectal-No-Chemo group (IR 9.6 [95% CI: 8.8–10.4] per 1000 person-years). Across all groups, the T2D IR was consistently higher in males than in females, in patients with BMI≥25 compared with BMI of 18.5–24.9, and in ASA>I compared with ASA score I (*Table 2*).

### Risk of developing T2D after different types of colorectal cancer surgery with and without chemotherapy

When compared with the Right-No-Chemo group, the other groups had similar and non-significant unadjusted HR of developing T2D (*Table 3*). In the adjusted analysis, Rectal-No-Chemo was associated with a lower risk of T2D (HR 0.86 [95% CI 0.75–0.98]) compared with Right-No-Chemo. In the sub-analysis comparing All Colon resected with All Rectal resected, the unadjusted and adjusted HR for developing T2D were significantly lower for Rectal resected. Moreover, adjusting for cancer stage did not affect the results (*Supplementary file 1*).

### Risk of developing T2D after radiation therapy among rectal resected patients

Radiation therapy in the rectal resected groups had no impact on the IR of T2D (*Table 2*), and the unadjusted/adjusted HR of developing T2D was non-significant when comparing Rectal-No-Radiation patients with Rectal-Radiation patients (*Table 3*).

### The impact of BMI on development of T2D

BMI had a strong association with T2D regardless of surgical resection and treatment with chemotherapy (*Figure 2*). The risk of developing T2D increased near twofold every time the BMI group increased one level, e.g., in the Right-No-Chemo group: HR 2.13 in BMI 25–30, HR 3.63 in BMI 30–35, HR 8.01 in BMI 35–40, compared with the reference BMI of 18.5–25. A similar pattern was observed within each of the other five groups, although a less pronounced impact of BMI was observed in the Left-No-Chemo group: HR 1.85 in 25–30, 2.93 in 30–35, 4.21 in 35–40 BMI group.

## Discussion

This national cohort study demonstrated an IR of developing T2D after CRC surgery similar to previous studies (*Singh et al., 2016*; *Jo et al., 2021*). Comparing resection types, rectal cancer resected patients had a slightly lower risk of developing T2D when compared to colon cancer resected patients. Treatment with chemotherapy had no impact on T2D risk in any of the resection groups, and radiation therapy did not affect T2D risk either.

Within each of the six groups (different types of surgery with and without chemotherapy), BMI was strongly associated to developing T2D, the risk increased almost twofold every time the BMI group increased one level.

Only few studies have examined the risk of developing T2D after different kinds of colonic and rectal surgical procedures, and in contrast to the present study the following studies were not able to adjust for several confounders incl. preoperative BMI. *Jensen et al., 2018* found a higher risk of T2D after the left-sided compared with the right-sided colonic resections, but, in contrast to the current study, patients operated for both non-malignant and malignant diseases were included. In support of side-specific effect of colonic resections, *Wu et al., 2021* investigated a population of non-CRC patients and showed that right-sided and transversal colonic resections were associated

**Table 1.** Baseline characteristics.

| | Right-sided resection (Right) (N=15,211) | | Left-sided resection (Left) (N=15,408) | | Rectum resection (Rectal) (N=15,754) | |
|---|---|---|---|---|---|---|
| | *Right-No-Chemo* | *Right-Chemo* | *Left-No-Chemo* | *Left-Chemo* | *Rectal-No-Chemo* | *Rectal-Chemo* |
| All patients, n (%) | 10,566 (23%) | 4,645 (10%) | 10,151 (22%) | 5,257 (11%) | 9,618 (21%) | 6,136 (13%) |
| Sex, n (%) | | | | | | |
| Male | 4,338 (41%) | 2,126 (46%) | 5,450 (54%) | 2,834 (54%) | 5,743 (60%) | 3,745 (61%) |
| Female | 6,228 (59%) | 2,519 (54%) | 4,701 (46%) | 2,423 (46%) | 3,875 (40%) | 2,391 (39%) |
| Age, year, mean (SD) | 75.2 (10.0) | 66.9 (10.1) | 72.1 (10.7) | 64.8 (10.1) | 70.0 (10.7) | 64.3 (10.1) |
| BMI, mean (SD) | 25.0 (4.7) | 25.4 (4.7) | 25.7 (4.6) | 25.8 (4.5) | 25.6 (4.4) | 25.6 (4.4) |
| BMI subgroups | | | | | | |
| <18.5 | 482 (4.6%) | 114 (2.5%) | 287 (2.8%) | 97 (1.8%) | 225 (2.3%) | 143 (2.3%) |
| 18.5–24.9 | 4,183 (40%) | 1,907 (41%) | 3,638 (36%) | 2,004 (38%) | 3,693 (38%) | 2,570 (42%) |
| 25–29.9 | 2,671 (25%) | 1,347 (29%) | 2,945 (29%) | 1,641 (31%) | 3,021 (31%) | 1,993 (32%) |
| 30–34.9 | 814 (7.7%) | 404 (8.7%) | 961 (9%) | 545 (10%) | 889 (9.2%) | 611 (10%) |
| 35–39.9 | 194 (1.8%) | 90 (1.9%) | 214 (2.1%) | 135 (2.6%) | 171 (1.8%) | 132 (2.2%) |
| >40 | 62 (0.6%) | 34 (0.7%) | 69 (0.7%) | 39 (0.7%) | 60 (0.6%) | 30 (0.5%) |
| Missing | 2,160 (20%) | 749 (16%) | 2,037 (20%) | 796 (15%) | 1,559 (16%) | 657 (11%) |
| Smoking, n (%) | | | | | | |
| Never | 3,132 (30%) | 1,533 (33%) | 3,030 (30%) | 1,804 (34%) | 2,828 (29%) | 2,014 (33%) |
| Current | 1,671 (16%) | 791 (17%) | 1,486 (15%) | 821 (16%) | 1,679 (17%) | 1,305 (21%) |
| Former | 3,376 (32%) | 1,465 (32%) | 3,415 (34%) | 1,700 (32%) | 3,421 (36%) | 2,021 (33%) |
| Missing | 2,387 (22%) | 856 (18%) | 2,220 (21%) | 932 (18%) | 1,690 (18%) | 796 (13%) |
| Alcohol*, n (%) | | | | | | |
| 1–14 | 4,819 (46%) | 2,342 (51%) | 4,803 (47%) | 2,771 (53%) | 4,886 (51%) | 3,399 (55%) |
| 14–21 | 478 (5%) | 254 (5%) | 618 (6%) | 333 (6%) | 711 (7%) | 494 (8%) |
| >21 | 433 (4%) | 256 (6%) | 593 (6%) | 323 (6%) | 671 (7%) | 412 (7%) |
| None | 2,269 (21%) | 889 (19%) | 1,722 (17%) | 872 (17%) | 1,446 (15%) | 1,006 (16%) |
| Missing | 2,567 (24%) | 895 (19%) | 2,415 (24%) | 958 (18%) | 1,904 (20%) | 825 (13%) |
| ASA score[†], n (%) | | | | | | |
| Healthy (=I) | 1,633 (15%) | 1,331 (29%) | 2,185 (22%) | 1,735 (33%) | 2,541 (26%) | 2,082 (34%) |
| Mild (=II) | 5,457 (52%) | 2,596 (56%) | 5,299 (51%) | 2,857 (54%) | 5,265 (55%) | 3,415 (56%) |
| Sick (=>II) | 3,160 (30%) | 613 (13%) | 2,442 (24%) | 548 (11%) | 1,653 (17%) | 557 (9%) |
| Missing | 316 (3%) | 105 (2%) | 295 (3%) | 117 (2%) | 159 (2%) | 82 (1%) |
| Screening detected tumor, n (%) | 770 (7%) | 299 (6%) | 1,077 (11%) | 457 (9%) | 664 (7%) | 394 (6%) |
| UICC stage[‡], n (%) | | | | | | |
| I/II | 7,002 (66%) | 674 (15%) | 7,339 (72%) | 745 (14%) | 5,940 (62%) | 420 (7%) |
| III/IV | 3,330 (32%) | 3,504 (75%) | 2,506 (25%) | 3,989 (76%) | 2,246 (23%) | 2,471 (40%) |
| Unknown | 234 (2%) | 467 (10%) | 306 (3%) | 523 (10%) | 1,432 (15%) | 3,245 (53%) |
| Radiation, n (%) | - | - | - | - | 1,157 (12%) | 2,983 (49%) |

Chemo: chemotherapy.

*Alcohol items per week.

[†]The American Society of Anesthesiologist score: I: healthy, but with CRC; III: mild systemic disease without substantial functional limitations; >II: severe systemic disease, includes ASA stages III, IV, V, VI.

[‡]UICC stage, Union of International Cancer Control; stage I: T1 or T2; stage II: T3 or T4; stage III: N1 or N2; stage IV: Disseminated disease at time of diagnosis.

**Table 2.** Absolute incidence rates of type 2 diabetes (T2D) per 1000 person-years (95% CI) among colorectal cancer patients treated with different types of colorectal cancer surgery with and without chemotherapy.

| | Right-sided resection (Right) (N=15,211) | | Left-sided resection (Left) (N=15,408) | | Rectum resection (Rectal) (N=15,754) | |
|---|---|---|---|---|---|---|
| | *Right-No-Chemo* | *Right-Chemo* | *Left-No-Chemo* | *Left-Chemo* | *Rectal-No-Chemo* | *Rectal-Chemo* |
| All patients, n | 10,566 | 4,645 | 10,151 | 5,257 | 9,618 | 6,136 |
| T2D development | | | | | | |
| Numbers | 495 | 235 | 609 | 323 | 560 | 334 |
| Mean time, year (SD) | 4.1 (3.4) | 4.1 (3.6) | 4.1 (3.5) | 4.4 (3.6) | 4.7 (3.7) | 4.3 (3.5) |
| Person-years | 48,039 | 22,317 | 53,935 | 29,301 | 58,448 | 33,422 |
| Incidence rates | 10.3 (9.4–11.2) | 10.5 (9.3–12.0) | 11.3 (10.4–12.2) | 11.0 (9.9–12.3) | 9.6 (8.8–10.4) | 10.0 (9.0–11.1) |
| Sex | | | | | | |
| Male | 13 (11–15) | 14 (12–16) | 14 (12–15) | 14 (12–16) | 11 (10–12) | 12 (11–14) |
| Female | 9 (8-10) | 8 (7-10) | 9 (8-10) | 8 (7-10) | 8 (7-9) | 7 (6-8) |
| Age | | | | | | |
| <50 | 3 (1-8) | 6 (3-11) | 7 (5-12) | 8 (5-12) | 5 (3-8) | 6 (4-10) |
| 50–64.9 | 12 (10–14) | 13 (11–16) | 11 (9–12) | 10 (8–12) | 10 (8–12) | 10 (9–12) |
| 65–74.9 | 12 (11–14) | 11 (9–14) | 12 (11–14) | 14 (12–17) | 14 (12–17) | 11 (10–13) |
| ≥75 | 9 (8-10) | 6 (4-9) | 11 (10–13) | 8 (6-12) | 8 (6-12) | 8 (5-11) |
| BMI subgroups | | | | | | |
| 18.5–24.9 | 6 (5-7) | 5 (4-7) | 6 (5-8) | 4 (3-5) | 4 (4-5) | 4 (3-6) |
| 25–29.9 | 13 (11–16) | 11 (9–14) | 13 (11–15) | 12 (10–15) | 11 (9–12) | 12 (10–14) |
| 30–34.9 | 23 (18–28) | 30 (23–40) | 21 (17–25) | 24 (19–30) | 25 (21–30) | 24 (19–30) |
| 35–39.9 | 51 (38–70) | 53 (34–84) | 32 (23–45) | 34 (23–52) | 30 (21–44) | 31 (20–48) |
| Missing | 7 (6-9) | 9 (6-12) | 11 (9–12) | 12 (10–16) | 9 (8-12) | 10 (8–14) |
| Smoking | | | | | | |
| Never | 9 (8-11) | 11 (9–14) | 9 (8-11) | 8 (7-10) | 8 (7-10) | 9 (7-11) |
| Current | 13 (10–15) | 9 (7-13) | 11 (9–14) | 12 (10–16) | 10 (8–12) | 10 (7–12) |
| Former | 11 (10–13) | 12 (10–15) | 13 (11–15) | 13 (11–15) | 11 (9–12) | 11 (9–13) |
| Missing | 8 (7-10) | 9 (6-12) | 11 (10–14) | 12 (9–15) | 9 (8-11) | 11 (8–14) |
| Alcohol** | | | | | | |
| 1–14 | 11 (9–12) | 10 (8–12) | 10 (9–12) | 10 (9–12) | 9 (8–10) | 10 (8–11) |
| 14–21 | 9 (6-14) | 7 (3-14) | 9 (6-13) | 10 (7–16) | 8 (6-11) | 9 (6-14) |
| >21 | 10 (7–16) | 16 (11–25) | 14 (11–19) | 12 (8–18) | 11 (9–15) | 13 (9–18) |
| None | 12 (10–14) | 14 (10–18) | 13 (11–16) | 12 (9–16) | 13 (11–15) | 10 (7–13) |
| Missing | 9 (7-10) | 10 (7–13) | 12 (10–14) | 12 (10–16) | 10 (9–10) | 10 (8–13) |
| ASA score†† | | | | | | |
| Healthy (=I) | 7 (6-9) | 5 (3-7) | 7 (6-9) | 7 (5-9) | 5 (4-6) | 6 (4-7) |
| Mild (=II) | 10 (9–12) | 12 (11–15) | 12 (11–13) | 13 (11–15) | 11 (10–12) | 13 (11–14) |
| Sick (=>II) | 14 (12–17) | 17 (13–24) | 17 (12–17) | 18 (14–25) | 15 (12–18) | 12 (8–17) |
| Missing | 6 (3-11) | 15 (7–31) | 13 (8–20) | 11 (5–23) | 9 (5-17) | 10 (4–25) |
| Radiation | | | | | | |
| Yes | - | - | - | - | 11 (9–13) | 11 (9–13) |
| No | - | - | - | - | 9 (9-10) | 10 (9–12) |

Chemo: chemotherapy.

*Alcohol items per week.

†The American Society of Anesthesiologist score: I: healthy, but with CRC; II: mild systemic disease without substantial functional limitations; >II: severe systemic disease, includes ASA stages III, IV, V, VI.

**Table 3.** Risk of developing type 2 diabetes after different types of colorectal cancer surgery with and without oncological treatment, unadjusted and adjusted analysis.

| | Unadjusted | | Model 1 | | Model 2[(a/b)] | |
|---|---|---|---|---|---|---|
| | HR (95% CI) | p-Value | HR (95% CI) | p-Value | HR (95% CI) | p-Value |
| Surgery and chemotherapy | | | | | | |
| Right-No-Chemo | ref | | | | | |
| Right-Chemo | 1.02 (0.87;1.19) | 0.799 | 0.99 (0.83;1.17) | 0.880 | 1.01 (0.85;1.20)[a] | 0.884 |
| Left-No-Chemo | 1.10 (0.98;1.24) | 0.112 | 0.92 (0.81;1.05) | 0.228 | 0.94 (0.82;1.07)[a] | 0.333 |
| Left-Chemo | 1.07 (0.93;1.23) | 0.339 | 0.93 (0.79;1.09) | 0.366 | 0.97 (0.83;1.14)[a] | 0.708 |
| Rectal-No-Chemo | 0.94 (0.83;1.06) | 0.317 | 0.82 (0.72;0.94) | 0.004 | 0.86 (0.75;0.98)[a] | 0.028 |
| Rectal-Chemo | 0.96 (0.84;1.11) | 0.610 | 0.84 (0.72;0.99) | 0.031 | 0.89 (0.76;1.03)[a] | 0.126 |
| Surgery and radiation therapy | | | | | | |
| Rectal-No-Radiation | ref | | | | | |
| Rectal-Radiation | 1.07 (0.93;1.24) | 0.336 | 0.98 (0.85;1.13) | 0.804 | 0.97 (0.84;1.12)[b] | 0.691 |
| Surgery | | | | | | |
| All Colon | ref | | | | | |
| All Rectal | 0.90 (0.83;0.98) | 0.013 | 0.87 (0.79;0.95) | 0.002 | 0.89 (0.82;0.98)[b] | 0.015 |

Right: right-sided colonic resections; Left: left-sided colonic resections; Rectal: rectal resections. No-Chemo: no chemotherapy; Chemo: chemotherapy. No-radiation: no radiotherapy; Radiation: radiation therapy. All Colon: all right- and left-sided colonic resections with and without chemotherapy. All Rectal: all rectal resections with and without chemotherapy, and with and without radiation therapy. Model 1: adjusted for year of surgery, age at surgery, sex, BMI. Model 2a: adjusted like Model 1 and further adjusted for performance status before surgery assessed by American Society of Anesthesiologist physical scale, smoking, and alcohol. Model 2b: adjusted like Model 2a and further adjusted for chemotherapy.

with a reduced risk of T2D development compared with patients without colectomy. In contrast to both the abovementioned studies the most recent epidemiological study found an increased T2D risk after colectomy compared with small bowel resection in non-cancer patients, but when stratifying by resection type, they found no difference in T2D risk (*Allin et al., 2022*).

In Denmark, the overall T2D prevalence (6.9%) (*Diabetesforeningen, 2023*), is lower than the global average in 2021 (10.5%) and also falls below the estimate of high-income countries (11.1%) (*Sun et al., 2022*). Similarly, the obesity rate of 20% aligns with other Scandinavian countries and is below that of most high-income nations (*Ritchie and Roser, 2023*).

Obesity is one of the predominant factors in the development of T2D (*Wang et al., 2005*; *Hu et al., 2014*; *Lee et al., 2018*), and the linear increase in new-onset T2D with increased BMI reported in this study is well described in the literature (*Easson Karin et al., 2022*). It is important to emphasise that only preoperative BMI was included in the adjustments as post-surgery BMI information was not available and accordingly, weight trajectories after surgery could not be included in the analyses. It's still unknown how body weight and body composition changes after different types of CRC resection, and thus it is not clear whether changes in weight in any groups play a role in diabetes development.

Like our findings, *Jensen et al., 2018* and *Wu et al., 2021* found a tendency towards a decreased risk of T2D after rectal resection, although these findings were not statistically significant in the previous studies. Further studies are needed to elucidate the potential mechanism behind this finding.

We found no impact of treatment with chemotherapy on T2D risk, which was a bit surprising. Prior studies indicated that treatment with chemotherapy may induce hyperglycemia, and especially the combined use of chemotherapy and glucocorticoid, administered to alleviate side effects of chemotherapy, may have synergistic effects on later diabetes risk (*Hwangbo et al., 2018*).

In the study period of the present study, different types of chemotherapy have been part of the standard treatment of colon and rectum cancer respectively, and knowledge regarding the specific

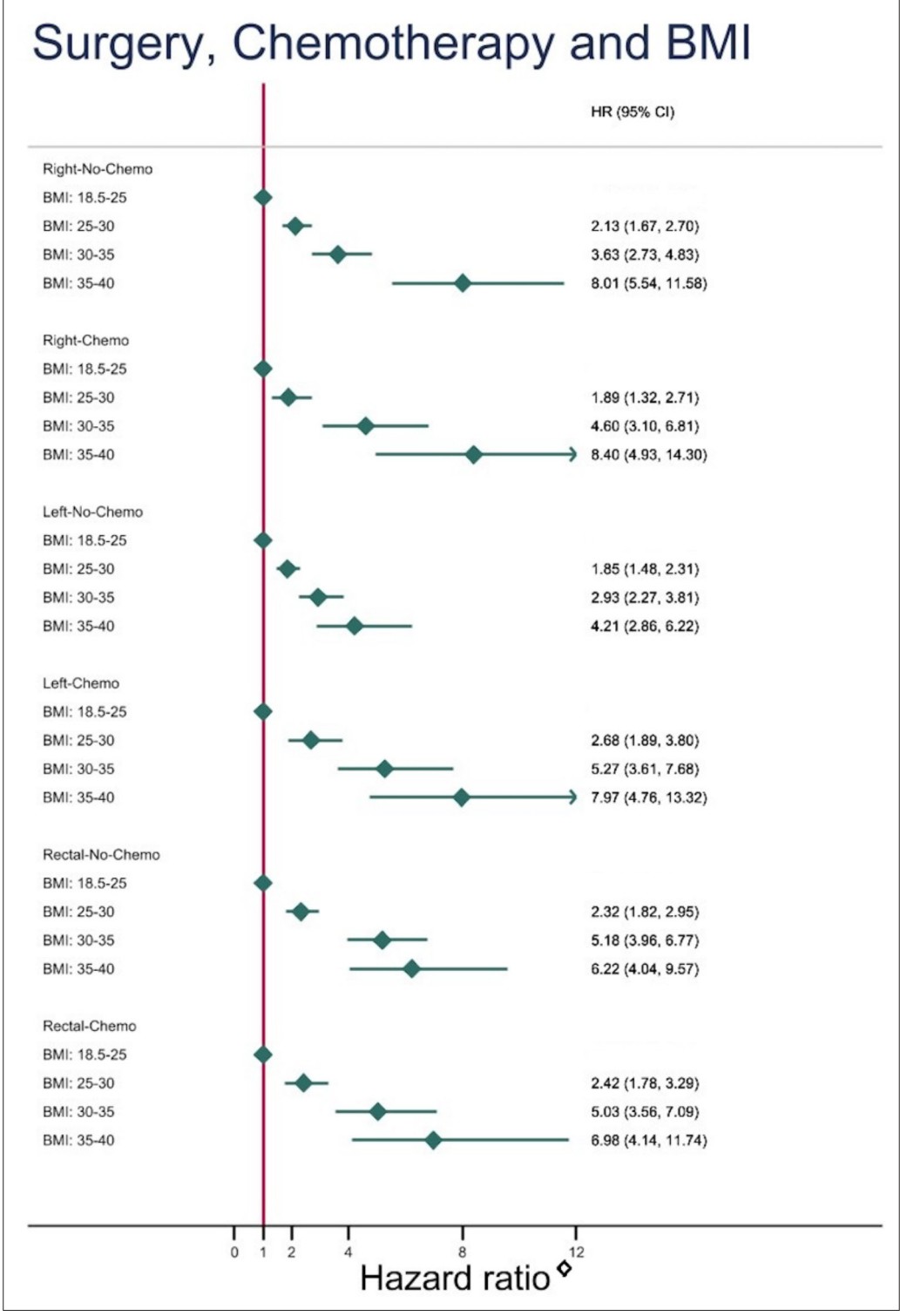

**Figure 2.** Risk of developing type 2 diabetes depending on body mass index (BMI) subgroups (using normal weight, BMI: 18.5–25 as reference) after different types of colorectal cancer surgery with and without chemotherapy.

effects of each drug on glucose homeostasis is sparse (*Ariaans et al., 2015*). In most cases CRC patients included in the study will however have been treated with the chemotherapy 5-fluorouracil (*Feng et al., 2013*). In a recent study a significant increase in glycemic variability, a precursor of T2D development, was found in non-diabetic early-stage colon cancer patients during 5-fluorouracil and capecitabine chemotherapy treatment (*Mandolfo et al., 2022*). Moreover, Feng et al. showed a 10% increased risk of developing T2D in CRC patients during and after treatment with adjuvant 5-fluorouracil. They argued that 5-fluorouracil could induce toxicity in B-cells leading to hyperglycemia (*Feng et al., 2013*). Similarly, *Lee et al., 2021* revealed an impairment in glucose homeostasis in both breast cancer and CRC patients treated with chemotherapy and glucocorticoid following treatment. Nonetheless 65% of the newly diagnosed T2D cases remitted 6 months after discontinuation of chemotherapy, suggesting that chemotherapy is a reversible diabetogenic factor (*Lee et al., 2021*). Thus, in summary, treatment with chemotherapy may primarily induce transient effects on glucose homeostasis in CRC patients. In contrast to the lack of influence of chemotherapy treatment, it was not surprising that radiation therapy had no impact on diabetes risk in our study.

Due to the strong negative prognostic effects of comorbidity with T2D on CRC outcomes, identifying CRC patients at increased risk of developing T2D is of great clinical importance. Our study did not support a role of cancer surgery with left-sided colonic resection or treatment with chemotherapy in T2D development in CRC survivors. However, in line with existing literature we found BMI to be strongly associated with T2D development.

A core strength of the present study is the use of a nationwide database, allowing inclusion of large numbers of CRC patients treated with different types of CRC resections, providing valid real-life data regarding with close to complete coverage and long-term follow-up and a register-based approach to identify individuals with T2D. We were able to adjust for several confounders such as BMI, alcohol consumption, and smoking, which has been a limitation in previous studies (*Jensen et al., 2018*; *Wu et al., 2021*).

The limitation of the present study includes that information on BMI and other risk factors for developing T2D were only available preoperatively, eliminating the possibility of exploring an impact of BMI trajectories after cancer treatment until the T2D diagnosis (*Winkels et al., 2016*). Secondly, information on dose, length of treatment, and type of chemotherapy were not available. Thirdly, information on physical activity, diet, medication, and comorbidities, apart from the ASA score, were not available. In addition, our findings are limited by the register-based data collection method. Clinical well-designed mechanistic investigations are warranted to establish whether left-sided CRC resection per se drives metabolic alterations leading to an increased risk of T2D.

In summary, we report a slightly increased risk of developing T2D after CRC treatment with colonic resection compared with rectal resection, whereas chemotherapy had no impact on new-onset T2D. Within each surgical subgroup, BMI at time of surgery was strongly associated to developing T2D. This study suggests that postoperative diabetes screening should be prioritised in CRC survivors with overweight/obesity regardless of type of CRC treatment applied.

## Acknowledgements

The authors thank the DCCG for access to data and Danish Cancer Society for support. This work was supported by the Danish Diabetes Association and the Danish Diabetes Academy, which is funded by the Novo Nordisk Foundation, grant number *NNF17SA0031406*. The Centre for Physical Activity Research (CFAS) is supported by TrygFonden (grants ID 101390, ID 20045, and ID 125132).

## Additional information

### Funding

| Funder | Grant reference number | Author |
| --- | --- | --- |
| Danish Diabetes Academy | NNF17SA0031406. | Louise Lang Lehrskov |
| Danish Diabetes Association | | Maria Saur Svane |

| Funder | Grant reference number | Author |
|---|---|---|
| Tryg Foundation | 101390 | Louise Lang Lehrskov |
| Tryg Foundation | 20045 | Louise Lang Lehrskov |
| Tryg Foundation | 125132 | Louise Lang Lehrskov |

The funders had no role in study design, data collection and interpretation, or the decision to submit the work for publication.

### Author contributions

Caroline Krag, Formal analysis, Writing – original draft, Writing – review and editing; Maria Saur Svane, Conceptualization, Resources, Supervision, Funding acquisition, Validation, Visualization, Methodology, Writing – original draft, Project administration, Writing – review and editing; Sten Madsbad, Conceptualization, Funding acquisition, Methodology, Writing – original draft, Writing – review and editing; Susanne Boel Graversen, Writing – original draft, Writing – review and editing; Jesper Frank Christensen, Thorkild IA Sørensen, Conceptualization, Methodology, Writing – original draft, Writing – review and editing; Louise Lang Lehrskov, Conceptualization, Resources, Data curation, Formal analysis, Supervision, Funding acquisition, Validation, Investigation, Visualization, Methodology, Writing – original draft, Project administration, Writing – review and editing; Tinne Laurberg, Conceptualization, Resources, Data curation, Software, Formal analysis, Supervision, Validation, Investigation, Visualization, Methodology, Writing – original draft, Project administration, Writing – review and editing

### Author ORCIDs

Caroline Krag ⓘ https://orcid.org/0009-0000-3447-5782
Maria Saur Svane ⓘ http://orcid.org/0000-0002-7345-4471
Sten Madsbad ⓘ http://orcid.org/0000-0002-5017-1815
Thorkild IA Sørensen ⓘ http://orcid.org/0000-0003-4821-430X
Louise Lang Lehrskov ⓘ http://orcid.org/0000-0002-1947-4252
Tinne Laurberg ⓘ https://orcid.org/0000-0002-7555-2665

### Ethics

According to Danish law and the Committee on Health Research Ethics in the Central Denmark Region, the study required no ethical approval because it was based on secondary use of register data for research purposes. This committee also waived patient consent for the use of register data. The study was approved by the Central Region Denmark (file no. 1-16-02-304-19).

Reviewer #1 (Public Review): https://doi.org/10.7554/eLife.89354.3.sa1
Author response https://doi.org/10.7554/eLife.89354.3.sa2

## Additional files

### Supplementary files

• Supplementary file 1. Risk of developing T2D after different types of colorectal cancer surgery with and without chemotherapy – adjusted for cancer stage, Model 1 and Model 2.
• MDAR checklist

### Data availability

According to European law (GDPR) data containing potentially identifying or sensitive patient information are restricted; our data involving clinical participants are not freely available in a public repository. The data are retrieved from the "The Danish Health Data Authority" (https://sundhedsdatastyrelsen.dk/da/english/health_data_and_registers), a part of the Ministry of the Interior and Health in Denmark and the application for access to data is administered by Research Service. Researchers can request existing data for data analyses. Please visit the online data overview for an extensive overview of the available data and Research Service's current output (https://sundhedsdatastyrelsen.dk/da/registre-og-services). Data are available upon request via the Research Service ( https://sundhedsdatastyrelsen.dk/da/english/health_data_and_registers/research_services/apply).

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
