## [Editor Report · eLife assessment]

This **valuable** study presents findings that suggest the need for postoperative type 2 diabetes screening and that this should be prioritized in colorectal cancer survivors with overweight/obesity regardless of the type of colorectal cancer treatment applied. The evidence supporting the claims of the authors is **solid** and the authors use a population-based cohort study including all Danish colorectal patients who had undergone colorectal cancer surgery between 2001-2018. The work will be of interest to medical biologists, endocrinologists and oncologists working on colorectal cancer.

---

## [Referee Report · Reviewer #1 (Public Review)]

Summary:

In this study, the authors set out to determine whether colorectal cancer surgery site (right, left, rectal) and chemotherapy impact the subsequent risk of developing T2DM in the Danish national health register.

Strengths:

- The research question is conceptually interesting

- The Danish national health register is a comprehensive health database

- The data analysis was thorough and appropriate

-The findings are interesting, and a little surprising that there was no impact of chemotherapy on the development of T2DM

- The authors have addressed my previous clarifications and questions.

- Regarding the generalizability of this study, as the authors discuss the prevalence of T2DM and obesity are lower in Denmark than in a number of other high income countries. Therefore, similar studies in other populations would be of interest.

- The study includes individuals who filled a prescription for diabetes medication, so likely includes some individuals with transient hyperglycemia/steroid induced diabetes during chemotherapy, rather than those with new onset longterm T2DM.

Overall, the authors achieved their aims, and the conclusions are supported by their results as reported.

The results are unlikely to significantly impact clinical practice or T2DM screening in this population, however are of interest to the community.

---

## [Author Response]

The following is the authors’ response to the original reviews.

**eLife assessment**
This study presents valuable findings on diabetogenic risk from colorectal cancer (CRC) treatment. The authors claim that postoperative screening for type 2 diabetes should be prioritized in CRC survivors with overweight/obesity, irrespective of the oncological treatment received. The evidence supporting the claims is solid but requires confirmation in different populations. These results have theoretical or practical implications and will be of interest to endocrinologists, oncologists, general practitioners, gastrointestinal surgeons, and policymakers working on CRC and diabetes.

Author response: We thank you for taking the time to provide constructive feedback on our manuscript and for the useful suggestions. We have provided a point-by-point response to each of the reviewers’ comments with clearly marked changes to the manuscript.

**Public reviews**

**Reviewer #1 (Public Review):**
Summary:In this study, the authors set out to determine whether colorectal cancer surgery site (right, left, rectal) and chemotherapy impact the subsequent risk of developing T2DM in the Danish national health register.Strengths:The research question is conceptually interestingThe Danish national health register is a comprehensive health databaseThe data analysis was thorough and appropriateThe findings are interesting, and a little surprising that there was no impact of chemotherapy on the development of T2DMWeaknesses:This is not a weakness as such, but in the discussion, I would consider adding some brief comment on the international generalizability of the findings - e.g. demographic make up of the Danish population health register and background rates of DM and obesity in this population with CRC compared to countries on other continents.

Author response: We agree that this information would be valuable. It has now been added in the Discussion section.

Changes in manuscript:"In Denmark, the overall T2D prevalence is 6.9%25, lower than the global average in 2021 (10.5%) and also falls below the estimate of high-income countries (11.1%).26 Similarly, the obesity rate of 20% aligns with other Scandinavian countries and is below that of most high-income nations.27” (Page 8, line 256-258)

A little more information would be helpful regarding how T2DM was diagnosed in the registry.

Author response: We have now added a more thorough explanation of how T2D was diagnosed in the Methods section.

Changes in manuscript: “Diabetes is defined as the second occurrence of any event across three types of inclusion events: (1) Diabetes diagnosed during hospitalisation (2) diabetes-specific services received at podiatrist (3) purchases of glucose lowering. Thus, if a patient developed transient T2D during chemotherapy treatment, it will only be an inclusion event if they purchase glucose lowering drugs. Individuals were classified as having T1D if they had received prescriptions for insulin combined with a diagnosis of type 1 from a medical hospital department. Otherwise, diabetes was classified as type 2.22” (Page 5, line 154-160)

If someone did develop transient hyperglycemia requiring DM medications during chemotherapy, would the investigators have been able to identify these people?

Author response: Yes, we have added a sentence in the Methods section.

Changes in manuscript: “Thus, if a patient developed transient T2D during chemotherapy treatment, it will only be an inclusion event if they purchase glucose lowering drugs.” (Page 5, line 156-158)

Would they have been classified as T2DM based on filling a prescription for DM meds for a period of time? Also, did the authors have information regarding time to development of T2DM after surgery?

Author response: Yes, if they have 2 (or more) prescriptions of oral glucose lowering drugs. Yes, we have information regarding time to development of T2DM after surgery and found no difference between the groups.

Changes in manuscript: Information on mean time to develop T2D post-surgery has now been added to Table 2.

In the adjusted Models, the authors did not adjust for cancer stage, even though cancer stage appears to be very different between the chemo and no chemo groups. It would be interesting to know if it affects the results if the model adjusted for cancer stage

Author response: We agree that adjustment for cancer stage would be a valuable information and we have performed the analysis and added a sentence in the Result section.

Changes in manuscript: An adjusted analysis of cancer stage now appears in the Supplementary table 1.

“Moreover, adjusting for cancer stage did not affect the results (Supplementary table 1).” (Page 7, line 219-220)

It would be worthwhile to report if mortality rates were different between the groups during follow up, and if the authors investigated whether perhaps differences in mortality rates led to specific groups living longer, and therefore having more time to develop DM

Author response: This situation is accounted for in the analysis by using Cox-regression analysis. This method accounts for the potential competing effect of mortality.

Changes in manuscript: None.

Overall, the authors achieved their aims, and the conclusions are supported by their results as reported.

The results are unlikely to significantly change patient treatment or T2DM screening in this population. With some additional information, as described above, the results would be of interest to the community.

**Reviewer #2 (Public Review):**
Summary:The study showed the impact of cancer treatment on new onset of diabetes among patients with colorectal cancer using the national database. Findings reported that individuals with rectal cancer without chemotherapy were less likely to develop diabetes but among other groups, treatment didn't show any impact on the development of diabetes. BMI still played a significant role in developing diabetes regardless of treatment types.Strengths:One of the strengths of this study is innovative findings about the prognosis of colorectal cancer treatment stratified by treatment types. Especially, as it examined the impact of treatment on the risk of new chronic disease after diagnosis, it became significant evidence that suggests practical insights in developing a proper monitoring system for patients with colorectal cancer and their outcomes after treatment and diagnosis. It is imperative for providers to guide patients and caregivers to prevent adverse outcomes like new onset of chronic disease based on BMI and types of treatment. The next strength is the national database. As the study used the national database, the generalizability is validated.Weaknesses:Even though the study attempted to examine the impact of each treatment option, the dosage of chemotherapy and the types of chemotherapy were not able to be examined due to the data source.

Author response: No unfortunately not. We agree that this would have been valuable information.This is stated in the original manuscript as a limitation. Please refer to page 10 line 305-306.

Changes in manuscript: None.

**Recommendations for the authors:**

**Reviewer #1 (Recommendations For The Authors):**
Minor things:There are minor inconsistencies in the methods and results regarding BMI. In the methods, the authors state that BMI <18.5 and >/=40 were excluded, but these groups are included in Table 2.

Author response: This has been corrected

Changes in manuscript: BMI groups <18.5 and >/=40 are now excluded in Table 2. (Page 18)

Line 204, I believe should be BMI 18.5-24.9, not 20-24.9.

Author response: This has been corrected

Changes in manuscript: “For each group (type of surgery ± chemotherapy), the HR for developing T2D depending on BMI subgroups was calculated by using Cox regression analysis adjusted for age, sex, year of surgery, and ASA score using normal weight (BMI:18.5-24.9) as the reference group.” (Page 6, line 184-186)

Rather than showing the BMI mean in Table 1, it would be interesting to see the BMI breakdown by category.

Author response: Yes, we agree. This analysis has now been added to Table 1

Changes in manuscript: Please refer to Table 1

Re line 215, I would consider rewriting to remove the multiple negatives -e.g. Radiation therapy in rectal resected had did not impact the incidence rate of T2D in the Rectal-No-Chemo group or Rectal-Chemo group

Author response: This has been corrected. Please refer to the Result section.

Changes in manuscript: “Radiation therapy in the rectal resected groups had no impact on the incidence rate of T2D (Table 2); and the unadjusted/adjusted HR of developing T2D was non-significant when comparing Rectal-No-Radiation patients with Rectal-Radiation patients (Table 3).” (Page 7, 223-225)

Consider changing some of the "didn't"s in the discussion to "did not"

Author response: This has been corrected.

Changes in manuscript: Revised and corrected throughout the discussion.

**Reviewer #2 (Recommendations For The Authors):**
Some points need to be clarified and improved.In the method, patients with Type 1 Diabetes were excluded in the baseline but some patients were diagnosed with Type 1 diabetes after treatment and they were included in your analysis. It is interesting to identify Type 1 Diabetes after the treatment as an outcome, do you think that this diagnosis is caused by the treatment? And incidence rate or other HRs did not seem to include Type 1 Diabetes as stated in the methods. Did you exclude every Type 1 diabetes? If not, It needs to give further explanation about this outcome since the mechanism of Type 1 Diabetes and Type 2 Diabetes is different.

Author response: This matter has now been clarified in the Methods section.

Changes in manuscript: “Additionally, individuals diagnosed with Type 1 diabetes (T1D) either before or after surgery were excluded, along with those diagnosed with T2D preoperatively or within the first 2 weeks postoperatively, as the last group probably represents patients with preoperatively unknown pre-existing prediabetes or diabetes.22” (Page 4, line: 125-128)

Despite limited existing findings, some studies actually reported the incidence rates of Type 2 Diabetes among patients with CRC (Singh S, Earle CC, Bae SJ, et al. Incidence of Diabetes in Colorectal Cancer Survivors. J Natl Cancer Inst. 2016;108(6):djv402. Published 2016 Feb 2. doi:10.1093/jnci/djv402; Khan NF, Mant D, Carpenter L, Forman D, Rose PW. Long-term health outcomes in a British cohort of breast, colorectal and prostate cancer survivors: a database study. Br J Cancer. 2011;105 Suppl 1(Suppl 1):S29-S37. doi:10.1038/bjc.2011.420; Jo A, Scarton L, O'Neal LJ, et al. New onset of type 2 diabetes as a complication after cancer diagnosis: A systematic review. Cancer Med. 2021;10(2):439-446. doi:10.1002/cam4.3666) whereas your study examined the impact of the different types of treatments.

Author response: Our findings of T2D rate among CRC patients are now commented on in discussion section, and the abovementioned studies are included as references.

Changes in manuscript: “This national cohort study demonstrated an IR of developing T2D after CRC surgery similar to previous studies.5,11” (Page 8, line 237-238)

To strengthen the presentation, some places should be revised.Line 216: it says that Table 1 showed no impact of radiation therapy on the incidence rate of T2D. However, either the interpretation or the table number seems wrong. Table 1 does not have this information. Correct this statement.Line 239: There are typo and incomplete sentence. Check the sentence and correct the sentence.Line 257-261: It may be a systematic issue to separate these two paragraphs. But two paragraphs seem related so make them one paragraph.

Author response: These suggested changes have been made. Regarding line 216 the paragraph has been adjusted to the following:

Changes in manuscript: “Radiation therapy in the rectal resected groups had no impact on the incidence rate of T2D (Table 2); and the unadjusted/adjusted HR of developing T2D was non-significant when comparing Rectal-No-Radiation patients with Rectal-Radiation patients (Table 3).” (Page 7, 223-225)

Reference

(1) Araghi M, Soerjomataram I, Jenkins M, et al. Global trends in colorectal cancer mortality: projections to the year 2035. Int J Cancer. 2019;144(12):2992-3000. doi:10.1002/ijc.32055

(2) Arnold M, Sierra MS, Laversanne M, Soerjomataram I, Jemal A, Bray F. Global patterns and trends in colorectal cancer incidence and mortality. Gut. 2017;66(4):683-691. doi:10.1136/gutjnl-2015-310912

(3) González N, Prieto I, del Puerto-Nevado L, et al. 2017 Update on the Relationship between Diabetes and Colorectal Cancer: Epidemiology, Potential Molecular Mechanisms and Therapeutic Implications. Vol 8.; 2017. www.impactjournals.com/oncotarget

(4) Mills KT, Bellows CF, Hoffman AE, Kelly TN, Gagliardi G. Diabetes mellitus and colorectal cancer prognosis: A meta-analysis. Dis Colon Rectum. 2013;56(11):1304-1319. doi:10.1097/DCR.0b013e3182a479f9

(5) Singh S, Earle CC, Bae SJ, et al. Incidence of Diabetes in Colorectal Cancer Survivors. J Natl Cancer Inst. 2016;108(6). doi:10.1093/jnci/djv402

(6) Xiao Y, Wang H, Tang Y, et al. Increased risk of diabetes in cancer survivors: a pooled analysis of 13 population-based cohort studies. ESMO Open. 2021;6(4). doi:10.1016/j.esmoop.2021.100218

(7) Colorectal D, Nordcan 2019. 5-Year Age-Standardised Relative Survival (%), Males and Females. Accessed September 12, 2022. “https://nordcan.iarc.fr/en/dataviz/survival?cancers=520&set_scale=0&sexes=1_2&populations=208”" has been copied into your clipboard

(8) Nano J, Dhana K, Asllanaj E, et al. Trajectories of BMI Before Diagnosis of Type 2 Diabetes: The Rotterdam Study. Obesity. 2020;28(6):1149-1156. doi:10.1002/oby.22802

(9) Maddatu J, Anderson-Baucum E, Evans-Molina C. Smoking and the risk of type 2 diabetes. Translational Research. 2017;184:101-107. doi:10.1016/j.trsl.2017.02.004

(10) Lega IC, Lipscombe LL. Review: Diabetes, Obesity, and Cancer-Pathophysiology and Clinical Implications. Endocr Rev. 2020;41(1). doi:10.1210/endrev/bnz014(11) Jo A, Scarton L, O’Neal LTJ, et al. New onset of type 2 diabetes as a complication after cancer diagnosis: A systematic review. Cancer Med. 2021;10(2):439-446. doi:10.1002/cam4.3666

(12) Feng JP, Yuan XL, Li M, et al. Secondary diabetes associated with 5-fluorouracil-based chemotherapy regimens in non-diabetic patients with colorectal cancer: Results from a single-centre cohort study. Colorectal Disease. 2013;15(1):27-33. doi:10.1111/j.1463-1318.2012.03097.x

(13) Lee EK, Koo B, Hwangbo Y, et al. Incidence and disease course of new-onset diabetes mellitus in breast and colorectal cancer patients undergoing chemotherapy: A prospective multicenter cohort study. Diabetes Res Clin Pract. 2021;174. doi:10.1016/j.diabres.2021.108751